# Electrodialysis Extraction of Pufferfish Skin (*Takifugu flavidus*): A Promising Source of Collagen

**DOI:** 10.3390/md17010025

**Published:** 2019-01-04

**Authors:** Junde Chen, Min Li, Ruizao Yi, Kaikai Bai, Guangyu Wang, Ran Tan, Shanshan Sun, Nuohua Xu

**Affiliations:** 1Marine Biological Resource Comprehensive Utilization Engineering Research Center of the State Oceanic Administration, The Third Institute of Oceanography of the State Oceanic Administration, Xiamen 361005, China; yiruizao@163.com (R.Y.); kkbai@tio.org.cn (K.B.); 17859733637@163.com (G.W.); tanran@tio.org.cn (R.T.); shshsun123@163.com (S.S.); xunuohua@163.com (N.X.); 2Plants for Human Health Institutes, North Carolina State University, Kannapolis, NC 28081, USA

**Keywords:** skin collagen, electrodialysis, thermal stability, *Takifugu flavidus*

## Abstract

Collagen is widely used in drugs, biomaterials, foods, and cosmetics. By-products of the fishing industry are rich sources of collagen, which can be used as an alternative to collagen traditionally harvested from land mammals. However, commercial applications of fish-based collagen are limited by the low efficiency, low productivity, and low sustainability of the extraction process. This study applied a new technique (electrodialysis) for the extraction of *Takifugu flavidus* skin collagen. We found electrodialysis to have better economic and environmental outcomes than traditional dialysis as it significantly reduced the purification time and wastewater (~95%) while maintaining high extraction yield (67.3 ± 1.3 g/100 g dry weight, *p* < 0.05). SDS-PAGE, amino acid composition analysis, and spectrophotometric characterization indicated that electrodialysis treatment retained the physicochemical properties of *T. flavidus* collagen. Heavy metals and tetrodotoxin analyses indicated the safety of *T. flavidus* collagen. Notably, the collagen had similar thermal stability to calf skin collagen, with the maximum transition temperature and denaturation temperature of 41.8 ± 0.35 and 28.4 ± 2.5 °C, respectively. All evidence suggests that electrodialysis is a promising technique for extracting collagen in the fishing industry and that *T. flavidus* skin collagen could serve as an alternative source of collagen to meet the increasing demand from consumers.

## 1. Introduction

Collagen is the major structural component of various extracellular matrices in mammalian connective tissues, such as the skin, corneas, cartilage, bone, and blood vessel [1]. Collagen is also widely used in biomaterials, drugs, foods, and cosmetics [2]. Its high biocompatibility as well as the ability to support cell growth and differentiation has made it an important matrix for cell biology, cosmetics, and regenerative medicine [3,4]. Collagen is also widely used as a gelatin precursor in the food industry for formulating emulsions, foams, colloids, and biodegradable films [5]. Annual sales of collagen and its derivatives in the global market have reached billions of dollars [6]. Despite high feasibility and biocompatibility, these mammalian origins potentially limit practical application of collagen due to sociocultural beliefs (e.g., in Muslim countries and India). In addition, the mammalian origins can further restrain collagen application by increasing additional sanitary costs for industrial production because of extensive consumer concerns regarding transmissible diseases from porcine and bovine [7,8].

Fish, a popular dietary ingredient, is a great alternative source of commercial collagen. In addition to its worldwide acceptance by different sociocultural belief systems, fish is less likely to carry mammalian-borne diseases, thus requiring lower sanitary costs for industrial production of collagen [9]. More interestingly, the rapid development of the fishing industry has resulted in a huge amount of collagen-rich by-products, including skin, scales, and bones (50.2–117 million metric tons) [10]. Pufferfish aquaculture is a thriving industry in China, producing more than 14,000 metric tons in 2013 [11]. Studies on fish suggest that pufferfish skin is rich in type I collagen and may serve as an appropriate source of collagen to replace their mammalian counterparts.

While promising, industrial production of fish collagen is limited by the low efficiency, low productivity, and low sustainability of the purification process. Crude fish collagen extract is mainly purified by dialysis through a passive diffusion process in the industry. This method usually takes 3–4 days to isolate pure collagen and often processes less than 1 L of samples [12,13]. Moreover, dialysis of crude fish collagen extract can produce a relatively large amount of acetified wastewater (20 L dialysate/L sample) [14,15,16], leading to severe environmental stress. An advanced purification technique with better economic and environmental outcome is therefore needed to take full advantage of collagen-rich fish by-products.

Electrodialysis, as an active diffusion technique, appears to be an excellent solution to improve the efficiency, productivity, and sustainability of fish collagen purification. This method can purify charged proteins/peptides by ion-exchange membranes through a stimulated diffusion process under the influence of electric potential difference [17]. A recent study on marine protein hydrolysate showed that electrodialysis could process 3 L of sample within a relative short time (6 h) while requiring just a small amount of dialysis buffer (1 L/L sample) [18].

The aim of this study was to assess the practicality of electrodialysis for the isolation and purification of collagen derived from yellowbelly pufferfish (*Takifugu flavidus*) skin. Considering that this technique has not been applied in the fishing industry, we hypothesized that the application of electrodialysis could not only improve economic and environmental outcomes of fish collagen production but also retain the physicochemical properties of *T. flavidus* collagen. Extraction efficiency, productivity, and wastewater production were also determined in this study, and the quality of *T. flavidus* collagen was systemically evaluated by electrophoresis, spectrophotometric characterization, thermal properties, and solubility.

## 2. Results and Discussion

### 2.1. Quality Measurements

Heavy metals can lead to chronic heart disease, cancer, and death [19]. Heavy metals have been banned by law from foods, cosmetics, and drugs [20]. The metal concentrations in pufferfish skin were found to be Pb (0.562 ± 0.052 mg kg^−1^), Cd (0.042 ± 0.004 mg kg^−1^), As (0.075 ± 0.002 mg kg^−1^), and Cr (1.152 ± 0.067 mg kg^−1^), which are below the permissible limit recommended by the Chinese national standards (Table 1) [20]. Some previous works have suggested that the accumulation of heavy metals in fish depends on the dietary ingestion rate and the concentration of heavy metals in the ingested food [21]. Thus, the low level of heavy metal concentration in pufferfish skin might be due to the artificial feed of lower heavy metals in aquaculture pufferfish. Moreover, the metal concentration in the collagen sample in this study was Pb (0.421 ± 0.015 mg kg^−1^), Cd (0.023 ± 0.002 mg kg^−1^), As (0.052 ± 0.001 mg kg^−1^), and Cr (0.863 ± 0.072 mg kg^−1^), which are also below the permissible limit recommended by the Chinese national standards (Table 1) [20].

The pufferfish (*T. flavidus*) skin contains 1.7 ± 0.5 MU/g of tetrodotoxin. According to Tani [22], the oral lethal dose of tetrodotoxin for a human being is 10,000 MU, and pufferfish with a tetrodotoxin content of less than 10 MU/g is considered to be nontoxic. Thus, pufferfish (*T. flavidus*) skin is safe for use in food, cosmetics, and pharmaceutical raw materials. Additionally, the content of tetrodotoxin in the collagen sample in this study was below the detection limit, indicating the safety of pufferfish collagen.

### 2.2. Sodium Dodecyl Sulfate Polyacrylamide Gel Electrophoresis (SDS-PAGE) Analyses of T. flavidus Collagen

The electrophoretic pattern of *T. flavidus* collagen was similar to the authentic standard of rat type І collagen (Figure 1), indicating an intact collagen profile after electrodialysis. It consisted of four major protein components with molecular weights of 122, 130, 250, and 310 kDa. The first two components had mass values close to α1 and α2 subunits of rat type I collagen. The ratio of 122/130 kDa components (~1:2) was on a level similar to rat collagen standard and consistent with a previous report on type I collagen extract of *T. rubripes* skin [23]. Therefore, 122 and 130 kDa proteins were identified as α2 and α1 subunits of type I collagen, respectively. On the other hand, the last two protein components (250 and 310 kDa) were tentatively identified as β subunit (dimer) and γ subunit (trimer) of type I collagen, respectively. These two proteins had counterparts with similar molecular weights in rat type I collagen standard (Figure 1) as well as other fish collagen [24]. They were possibly formed via intermolecular and/or intramolecular cross-linking of collagen subunits [25]. As starvation is believed to stimulate collagen cross-linking of fish, the difference in β and γ subunit contents between *T. flavidus* and *T. rubripes* collagen were probably due to variations in both species and feeding conditions [23]. In addition, quantification of stained protein bands showed that α1, α2, β, and γ subunits contributed to 96.1 ± 1.3% of total *T. flavidus* collagen. This indicated that extraction by salting-out electrodialysis was able to produce pure type I collagen for further application.

### 2.3. Amino Acid Composition of T. flavidus Collagen

*T. flavidus* collagen demonstrated the characteristic amino acid composition of type I collagen. Gly was the most abundant residue of the collagen, accounting for a quarter of the total amino acid components (Table 2, 268 ± 0.08 residues/1000 amino acid residues). This observation is consistent with the common understanding that Gly content is the highest among all amino acid residues as type I collagen is featured with repeating Gly–Pro–X and/or Gly–X–Hyp sequence (X can be any amino acid residues other than Gly, Pro, and Hyp) [26].

Imino acids (Pro and Hyp) had the second highest amino acid content among all residues of *T. flavidus* collagen (246 ± 0.04 residues/1000 amino acid residues), consistent with a previous study on *T. rubripes* skin collagen (170 residues/1000 amino acid residues) [2,23]. They have been reported to decrease the entropic cost of collagen folding by preorganizing individual poly-Pro II chain [27]. They can also stabilize collagen triple helices via interchain hydrogen bond through hydroxyl groups [28]. The content of imino acids is therefore an important factor modulating collagen thermal stability [27,29]. The imino acid content of *T. flavidus* collagen was higher than many fishes, including big eye snapper, grass carp, and tiger pufferfish (167–195 residues/1000 amino acid residues), but similar to those from their porcine counterparts (220 residues/1000 amino acid residues) [16,24,30]. This suggests that *T. flavidus* collagen might have nutritional values similar to those of common mammalian collagen and could therefore potentially be used as an alternative source of gelatin.

### 2.4. Spectrophotometric Characterization

The spectrophotometric characterization was in agreement with SDS-PAGE and amino acid composition analyses, further confirming that electrodialysis maintained the physicochemical integrity of type I collagen from *T. flavidus* skin. In Figure 2A, it can be seen that *T. flavidus* collagen and authentic type І collagen standard both have bell shape UV spectra, with the maximum absorption wavelength around 234 nm. This strong absorption can be attributed to peptide bond absorptions by n → π* transitions within C=O, COOH, and CONH_2_ groups of collagen peptide chains [25]. Similar to type I collagen of rat tail, *T. flavidus* collagen also absorbed weakly around 250 and 280 nm. The inability to absorb at higher UV regions is related to the deficiency of Tyr and Phe in *T. flavidus* collagen (<30 residues/1000 residues, Table 2) because Tyr and Phe are the major chromophores responsible for absorption at 251 and 276 nm [31]. This phenomenon suggests a possible deprivation of protease-labile telopeptides from type I collagen [31] during isolation and purification with salting-out electrodialysis extraction. In addition, circular dichroism (CD) analysis was in line with UV results, showing identical spectra between *T. flavidus* collagen and rat type I collagen standard (Figure 2B). Both samples had spectra with a positive amplitude at 221 nm and a negative amplitude at 197 nm. It is also in agreement with recent research on scale collagen of pacific saury [32].

Consistent with UV analysis, *T. flavidus* collagen demonstrated a typical Fourier transform infrared (FTIR) spectrum of type I collagen. Five characteristic peaks were identified in both *T. flavidus* collagen and rat type I collagen: amide A, B, I, II, and III (Figure 2C). The wavenumber of amide A (3311 cm^−1^) was lower than that of free N–H stretching vibration (3400–3440 cm^−1^) [33]. This red-shift is in agreement with the extensive distributions of N–H(Gly∙∙∙O=C(X) hydrogen bonds among *T. flavidus* collagen helices during the formation of triple-helical structures [34]. Amide B reflected asymmetrical CH_2_ stretching within collagen peptides [35], and it had a wavenumber (2926 cm^−1^) similar to theoretical values. As an indicator of C=O stretching vibration [33], amide I of *T. flavidus* collagen (1645 cm^−1^) was modestly red-shifted from the calculated value (~1660 cm^−1^) toward lower wavenumber [33]. This was again a by-product of intercollagen cross-linking by N–H(Gly)∙∙∙O=C(X) hydrogen bonds [34]. N–H bending coupled with C–N stretching vibration actively contributed to amide II formation, inducing FTIR absorption at 1550–1600 cm^−1^ [35]. The amide II wavenumber of *T. flavidus* collagen (1551 cm^−1^) was at the lower region of this range, further confirming the influence of interhelical hydrogen bonds. In addition, C–H stretching and N–H bending vibrations were detected in *T. flavidus* collagen [36] with evidence on amide III (1242 cm^−1^) absorption.

### 2.5. Thermal Properties

The thermal stability of *T. flavidus* collagen was characterized by denaturation temperature (T_d_) and maximum transition temperature (T_max_). T_d_ refers to the temperature at which the triple-helical collagen structure in solution is disintegrated into random coil [16]. The viscosity of *T. flavidus* collagen (T_d_ = 28.4 ± 2.5 °C, Figure 3A) was found to decrease modestly slower than that of Alaska pollack (T_d_ = 16.8 °C) [37], indicating a milder denaturing process for *T. flavidus* collagen. Furthermore, T_max_ refers to the temperature at which collagen fiber shrinks to one third of its length. The phase transition involving the conversion of a crystalline triple-helical collagen structure to an amorphous random coil form occurs during the shrinkage process [38]. Differential scanning calorimetry (DSC) analysis of *T. flavidus* collagen confirmed this observation with a T_max_ value of 41.8 ± 0.35 °C (Figure 3B). The difference between the T_max_ and T_d_ values of *T. flavidus* collagen was about 13 °C. The T_max_ value of *T. flavidus* collagen was higher than those of bighead carp, bigeye snapper, and grass carp (24.6–33.3 °C) [14,16,24]. It is therefore reasonable to speculate that *T. flavidus* collagen tri-helices are more stable at higher temperature than other fishes. More interestingly, the T_max_ value of *T. flavidus* collagen was similar to calf skin collagen (40.8 °C) [39]. Considering the importance of thermal stability for collagen applications in foods [5] and the aquacultural potential of *T. flavidus* [40], we postulate that collagen from *T. flavidus* could potentially be used as an alternative source of collagen.

The superior thermal stability of *T. flavidus* collagen was likely due to its imino acid content. Imino acids, especially Hyp, have been known for their ability to stabilize collagen tri-helices via intermolecular hydrogen bonds [41]. Our study showed that *T. flavidus* collagen (Table 2, 246 ± 0.04 residues/1000 amino acid residues) had significantly higher imino acid content than the skin collagen of bighead carp, bigeye snapper, grass carp, and *T. rubripes* (*p* < 0.05, 165–193 residues/1000 amino acid residues) [14,16,23,24]. The additional imino acids might form extra hydrogen bonds within *T. flavidus* collagen tri-helices, therefore increasing the molecular stability through rising entropy [42].

In addition, a secondary exothermal peak (32.9 ± 0.31 °C) was identified in the DSC thermogram (Figure 3B). It was consistent with our observation on the viscosity changes in *T. flavidus* collagen (Figure 3A), indicating a partial denaturation of collagen supramolecular structure due to defibration of thermally unstable hydroxyproline-free sequence in collagen triple helices [28].

### 2.6. Relative Solubility

In the absence of NaCl, 0.5 mol/L acetic acid was able to fully dissolve *T. flavidus* collagen at a concentration of 0.16 mg/mL. Increments in NaCl level (≤3 g/100 mL) appeared to have a modest impact on the collagen solution, leading to a minor reduction in the relative solubility (99.5%–89.8%, Figure 4A). Higher levels of NaCl (≥4 g/100 mL), however, significantly decreased the relative solubility (11.0%–11.7%, *p* < 0.05). This phenomenon is in agreement with an earlier study on catfish skin collagen, which significantly precipitates at high NaCl levels (≥4 g/100 mL) [43]. Considering that Na+ and Cl− are the major forces depriving water molecules from hydrophilic amino acid residues of collagen during NaCl-mediated salting-out process [44], it is reasonable to postulate that collagen solubility is affected by the ionic strength of solutions. To avoid collagen aggregation due to potential salting-out events, the ionic strength of 1.01 (equivalent to the solution containing 3 g/100 mL NaCl and 0.5 mol/L acetic acid) could serve as a threshold value for preparing *T. flavidus* collagen in other salt solutions.

The protein content of *T. flavidus* collagen at pH 3.0 (0.04 mg BSAE/mL) was used to determine the relative solubility of the collagen among all pH conditions (Figure 4B). While adjusting pH values within acidic environments could modestly reduce relative solubility to 82.1 ± 0.1% (1 ≤ pH ≤ 3.0, *p* < 0.05), sharper reductions were observed between pH 3.0 and 5.0, with the lowest relative solubility being 37.9 ± 0.01%. This event was also reported by Singh and colleagues in a research involving catfish skin collagen [43]. As ionic strengths of *T. flavidus* collagen solutions (0.50–0.65) were much lower than the threshold value (1.01) during pH adjustments, it might have had a minor impact on the relative solubility of the collagen.

The reduction in relative solubility of *T. flavidus* collagen between pH 3.0 and 5.0 was most likely due to deprotonation of the charged amino acid residues. In acidic environments (pH ≤ 3.0), collagen behaved as a positively charged particle, with most charged amino acid protonated. The net positive charges of collagen assisted in forming hydrogen bonds with water molecules and preventing aggregation with surrounding proteins [45]. However, when the solution pH was increased from 3.0 to 5.0, the collagen lost a large fraction of net positive charges due to deprotonation of the side chains within Asp (pK 3.86) and Glu (pKa 4.25), which accounted for 49.2% of the charged amino acid residues of the collagen (Table 2). Considering the increasing hydrophobic interactions among neutralized collagens [44], it is reasonable to postulate a sharp reduction in relative solubility.

In neutral or basic environments (pH ≥ 6), the relative solubility of *T. flavidus* collagen continuously decreased, with the lowest value found at pH 7.0 (22.7 ± 0.02%, Figure 4B). This observation is in accordance with previous studies on type I collagen [12]. Changes in relative solubility were partially due to the elevated ability of imino acids (Hyp in particular) to facilitate intermolecular cross-linking within collagen tri-helical structures [41,42] as imino acids contributed to 24.6% of the total amino acid compositions of *T. flavidus* collagen (Table 2). It could also be attributed to alterations in total net charges of collagen molecules as electrostatic expulsion is too weak to shield proteins from aggregation around the isoelectric point (pI ≈ 7 for type I collagen) [45].

### 2.7. Cell Proliferation

The greater the optical density (OD) value indicted, the greater is the number of cells. As shown in Figure 5, the OD value of the negative control and experimental group increased with culture time. However, at different points in time, the OD values of the experimental groups were slightly higher than those of the negative control group, although the difference was not significant. This may be because the cells were cultured at 37 °C, which was above the denaturation temperature of 28.4 ± 2.5 °C for the collagen samples. When the collagen sample was kept at 37 °C for a long time, the collagen degraded, leading to loss of collagen activity. In the next stage of research, one important aim of our research will be to improve the thermal stability of collagen.

### 2.8. Extraction of T. flavidus Collagen

To our knowledge, this is the first time electrodialysis was introduced as part of fish collagen purification. The application of electrodialysis significantly increased the extraction yield of fish collagen. Our experiments showed that 67.3 ± 1.3 g of *T. flavidus* collagen could be produced from 100 g of dry skin, which is much higher than the yields of *T. rubripes* collagen (10.7–44.7 g/100 g dried weight, DW) extracted by traditional dialysis [23,30]. This discernible difference could be partially attributed to the variations in collagen contents between the two *Takifugu* species. It is also likely due to electrodialysis-mediated improvements of extraction efficiency as the extraction yield obtained from this study was significantly higher than those reported by previous dialysis research on tilapia (27.2 g/100 g DW) [46] and grass carp (46.6 g/100 g DW) [16].

This study showed that electrodialysis produced more than 10 kg of collagen samples in one purification effort by purifying 100 L of crude collagen solution. It also concentrated the purification step from a traditional 96-h dialysis [13,14,15] into a 2-h period. Considering that the traditional method could not purify 100 L of crude collagen solution in one effort [12,13,14], the application of electrodialysis is estimated to significantly reduce instrument and labor time for large-scale production. This demonstrates that electrodialysis has a huge advantage over traditional dialysis for translation into industrial use.

In addition to improving purification efficiency and productivity, electrodialysis is a cost-effective technique with better environmental sustainability relative to the traditional method. In our study, to produce 10 kg of *T. flavidus* collagen, electrodialysis consumed 45.0 L of acetic acid and produced 600 L of waste water (Table 3). In comparison, traditional dialysis is estimated to use 400 times more acetic acid and 600 m of additional dialysis bag [12,14,15] while producing 600 times more waste water to purify the same amount of collagen extract. In total, it would take US$15 to produce 10 kg of collagen using electrodialysis extraction. This indicates a strong potential of the electrodialysis technique to be used in industrial applications.

## 3. Materials and Methods

### 3.1. Materials and Chemicals

*T. flavidus* skin was purchased from the local fishery factory (Zhangzhou, China). Upon arrival, the skin was washed with chilled distilled water and then stored at −20 °C until use. Coomassie blue R-250, *N,N,N′,N′*-tetramethylethylenediamime and sodium dodecyl sulfate were purchased from Bio-Rad Laboratories (Hercules, CA, USA). Type I collagen (C3867, rat) was purchased from Sigma-Aldrich Inc. (St. Louis, MO, USA). Electrophoresis loading buffer was obtained from Sangon Biotech Co., Ltd. (Shanghai, China). Other reagents (analytical grade) were purchased from Xiamen Green Reagent Glass Instrument Co. Ltd. (Xiamen, China).

### 3.2. Isolation and Purification of T. flavidus Collagen

*T. flavidus* skin (15 kg in dry weight, DW) was incubated in 150 L of 0.1 mol/L sodium bicarbonate for 3 h with continuous stirring and then rinsed with cold water until a neutral pH was reached. The resulting material was hydrolyzed in 120 L of 0.5 mol/L acetic acid for 24 h and then centrifuged at 10,000× *g* for 15 min to remove non-collagenous proteins and pigments. Skin collagen was salted out from the supernatant in the presence of 0.5 mol/L NaCl, followed by 20 min of 10,000× *g* centrifugation. The resulting precipitate was dissolved in 0.5 mol/L acetic acid to produce 100 L of crude collagen solution, which was then purified by electrodialysis. The samples were loaded to a DSA-П electrodialyzer (Jiangsu Ritai Environmental Protection Engineering Co., Ltd., Yangzhou, China) equipped with both polyethylene cation-exchange membrane (361BW, Jiangsu Ritai Environmental Protection Engineering Co., Ltd., China) and polyethylene anion-exchange membrane (362BW, Jiangsu Ritai Environmental Protection Engineering Co., Ltd., China). The process conditions of electrodialysis were as follows: feed compartment, 150 L of 3% NaCl (*w*/*v*); concentrate compartment, 150 L of distilled water; electric potential, 80 V/cm; flow rate, ≤1 m^3^/h; electrodialysis duration, 2 h. The purification products were lyophilized and stored at −20 °C until analysis. Isolation and purification were all performed at 4 °C to prevent microbial growth (Figure 6). The extraction yield was calculated based on the following equation.
Yield (g/100 g DW) = Mass of lyophilized collagen (g)/Mass of dry fish skin (100 g)(1)

### 3.3. Quality Measurements

#### 3.3.1. Heavy Metal

Toxic metals in the pufferfish skin and collagen sample were characterized based on a method described by Kosker et al. [47] with slight modification. Smashed pufferfish skin and collagen sample were dissolved in nitric acid (25% *v*/*v*) and then digested using a Milestone ETHOS A microwave digestion instrument (Milestone Srl., Sorisole, Italy). Inductively coupled plasma mass spectrometry (ICP-MS, Thermo Fisher X Series II, Thermo Fisher Scientific Inc., Waltham, MA, USA) was used to identify these metals. Standard solutions for the calibration curve were prepared according to the dilutions of the toxic metals. Prepared solutions of toxic metals had levels of Pb, Cd, As, and Cr within the range of 1–50 mg/L.

#### 3.3.2. Tetrodotoxin

Tetrodotoxin in the pufferfish skin and collagen sample was characterized based on a method described by Wang et al. [48] with slight modification. Smashed pufferfish skin and collagen sample were incubated in 0.5 mol/L acetic acid at 100 °C for 40 min and then centrifuged at 10,000× *g* for 15 min. The resulting supernatants were collected and analyzed directly with an enzyme-linked immunosorbent assay (ELISA) test kit (Beijing Zhongnuo Taian Technology Co. Ltd., Beijing, China) for tetrodotoxin. As described in the manual, samples and enzyme-labeled MAb-TTX were added to wells of a microplate in which standard tetrodotoxin had been immobilized as a competitive agent. The absorbance (OD value) was measured at 490 nm. The ratio of absorbance (Ai/Ao, where Ai is the absorbance of standard solution, and Ao is the absorbance of blank with no tetrodotoxin added) was used as the ordinate. The logarithm of the corresponding standard solution concentration was plotted as the abscissa, a standard curve was prepared, and the tetrodotoxin content of the sample was calculated.

### 3.4. SDS-PAGE

SDS-PAGE was conducted as described in our earlier study [2]. Briefly, 300 µL of 2 µg/µL *T. flavidus* extract was mixed with 100 µL of loading buffer, incubated in boiling water for 3 min, cooled at room temperature, and then centrifuged at 8500× *g* for 5 min. Supernatant (5 µL/lane) was characterized by a polyacrylamide gel (4% stacking and 8% running) on a mini vertical electrophoresis system (Bio-Rad Laboratories, US). After electrophoresis, the gel was incubated in fixing solution (methanol:acetic acid:water, 50:10:40) for 30 min, stained by 0.1% Coomassie brilliant blue for 30 min, and then rinsed by 30% methanol containing 10% acetic acid for 30 min. Contents of the collagen subunits were estimated by Quantity One 4.6.0 (Bio-Rad Laboratories, US).

### 3.5. Amino Acid Composition

Amino acid composition of sample was conducted as described in our earlier study [2]. The *T. flavidus* collagen extract (5 mg) was hydrolyzed with 0.50 mL of 6.0 M hydrochloric acid at 110 °C for 8 h. The hydrolysate was vaporized by a vacuum evaporator, dissolved in 25 mL 0.1 M HCl, and then analyzed by applying the hydrolysate to a HITACHI 835-50 amino acid analyzer (Hitachi, Tokyo, Japan). Contents of the amino acid residues were expressed as residues/1000 residues.

### 3.6. Spectrophotometric Characterization

Spectra of UV, FTIR, and CD of the purified *T. flavidus* collagen were determined using a method described by Wang et al. [49] with slight modification. UV spectrum (226–300 nm) of the *T. flavidus* collagen extract (1 mg/mL) was determined by a UV-1780 spectrophotometer against 0.5 M acetic acid (blank). FTIR (500–4000 cm^−1^) of the same extract was measured by a horizontal ATR trough plate crystal cell (PIKE technology Inc., Fitchburg, WI, USA) coupled with a Bruker Model VERTEX 70 FTIR spectrometer (Bruker Co., Karlsruhe, Germany). FTIR spectra in the rage of 500–4000 cm^−1^ with automatic signal gain were collected in 16 scans at a resolution of 4 cm^−1^ prior to analysis by OPUS 6.5 data collection software program (Bruker Co., Karlsruhe, Germany). In addition, CD spectrum (190–260 nm) of the collagen extract (0.25 mg/mL) was produced by accumulating 100 scans at a speed of 100 nm/min with 1 nm interval on a Chirascan spectrometer (Applied Photophysics Limited Inc., Leatherhead, UK).

### 3.7. Relative Solubility

The effects of pH and NaCl on sample solubility were characterized as described in our earlier study [46].

The relative solubility of the *T. flavidus* collagen extract was determined in the presence of 0–6 g/100 mL NaCl. Briefly, 8 mL of 3 mg/mL collagen extract was mixed with 5 mL of NaCl solution containing 0.5 M acetic acid. The mixture was centrifuged at 20,000× *g* and 4 °C for 30 min. The protein concentration of the supernatant was measured according to the calibration curve for bovine serum albumin by Folin assay [50]. The absolute protein concentration of the collagen extract was expressed as mg bovine serum albumin equivalent/mL (mg BSAE/mL). The relative solubility was calculated with Equation (2) below by normalizing the protein concentrations collected at all NaCl levels with values obtained from collagen extracts that received no NaCl treatment.
Relative solubility (%) = (Protein concentration of supernatant with NaCl treatment)/(Protein concentration of supernatant without NaCl treatment) × 100(2)

The relative solubility of the *T. flavidus* collagen extract was also determined at varied pH values (1–10). The collagen extract (8 mL, 3 mg/mL) was adjusted to designed pH condition by 6.0 M HCl (or NaOH) with a final volume of 10 mL. The resulting solution was centrifuged at 20,000× *g* and 4 °C for 30 min. The protein concentration of the supernatant was determined by Folin assay as described above. The relative solubility was calculated with Equation (3) below by normalizing the protein concentrations collected at pH conditions with values obtained at pH 3.0.
Relative solubility (%) = (Protein concentration of supernatant at varied pH)/(Protein concentration of supernatant at pH 3.0) × 100(3)

### 3.8. DSC

The thermal properties of the sample were characterized by an established DSC method [14] with slight modification. Lyophilized collagen extract was hydrated with 0.05 M acetic acid (1:40 *w*/*v*), incubated at 4 °C for 2 days and then characterized using a DSC 2 calorimeter (Mettler-Toledo, Zurich, Switzerland). Heat flow of the hydrated samples was measured between 5 and 75 °C at 1 °C/min using an empty aluminum pan as reference. T_max_ was defined as the peak of the transition curve.

### 3.9. Viscosity

The viscosity of the sample was characterized based on a method described by Kittiphattanabawon et al. [24] with slight modification. Briefly, protein dispersion (0.1% *w*/*v*) was prepared by hydrating the collagen sample in 0.05 M acetic acid at 4 °C for 2 days. An Ostwald’s viscometer (Kusano Inc., Tokyo, Japan) loaded with collagen dispersion was incubated in a 15 °C water bath for an extended period. Small temperature increments (2 °C) were then applied to the viscometer at 30-min intervals in a stepwise manner. Viscosity was recorded between 15 and 50 °C prior to each temperature change. Fractional viscosity was calculated with Equation (4). T_d_ was defined as the temperature for fractional viscosity to reach 0.50.
Fractional viscosity = (Measured viscosity − minimum viscosity)/(Maximum viscosity − minimum viscosity)(4)

### 3.10. Cell Proliferation

Cell proliferation experiments were performed using a method described by Golser et al. [51] with slight modification. The collagen sample was dissolved in 0.5 mol/L acetic acid and cast into cell culture wells at a final concentration of 0.5 mg/mL. After the collagen was fully dried, the plates were sterilized via UV treatment and seeded with the NIH3T3 fibroblasts. NIH3T3 fibroblasts seeded in the plates without collagen served as a negative control. After adding Dulbecco’s Modified Eagle Medium (DMEM) cell culture medium to the plates, the cells were cultured in an incubator at 37 °C. At cell seeded times of 1, 2, 3, 4, and 5 days, 3-(4,5-dimethylthiazol-2-y)-2,5-diphenyltetrazolium bromide (MTT) solution was added to the cell culture medium and incubated for 4 h at 37 °C. The plates were removed, and the culture medium was aspirated. After adding dimethyl sulfoxide (DMSO) for 10 min, the absorbance (OD value) was measured at 490 nm.

### 3.11. Statistical Analyses

All data in this study are expressed as average ± standard deviation of at least three measurements. One-way ANOVAs coupled with Dunnett’s test was performed using SPSS 13.0 to determine the statistical difference at 95% confidence level.

## 4. Conclusions

Our study, for the first time, introduced electrodialysis for the extraction of fish skin collagen (*T. flavidus*). This cost-effective technique was found to be superior to traditional dialysis with its advanced efficiency (2 h/trial), large capacity (100 L/trial), high extraction yield (67.3 ± 1.3 g/100 g DW), and better environmental sustainability (6 L waste water/L sample). SDS-PAGE, amino acid composition analysis, and spectrophotometric characterization demonstrated that *T. flavidus* collagen extracted by electrodialysis primarily consisted of type I collagen with a purity of 96.1 ± 1.3%. ICP-MS analysis demonstrated that the heavy metals of *T. flavidus* collagen were less than the Chinese national standards. ELISA analysis indicated the safety of *T. flavidus* collagen. Notably, the collagen appeared to have better thermal stability than other fish species, with T_max_ and T_d_ of 41.8 ± 0.35 and 28.4 ± 2.5 °C, respectively. This observation can be attributed to its relative higher imino acid content (246 ± 0.04 residues/1000 amino acid residues). Cell proliferation experiment demonstrated that the cell proliferation rates of the experimental groups were slightly higher than those of the negative control group, but there was no significant difference. In addition, both NaCl level and pH condition were found to affect the relative solubility of *T. flavidus* collagen. All evidence suggests that electrodialysis is a promising technique for fish collagen and that *T. flavidus* skin collagen could serve as an alternative source of collagen to meet the increasing demand from academia and industry.

## Figures and Tables

**Figure 1 marinedrugs-17-00025-f001:**
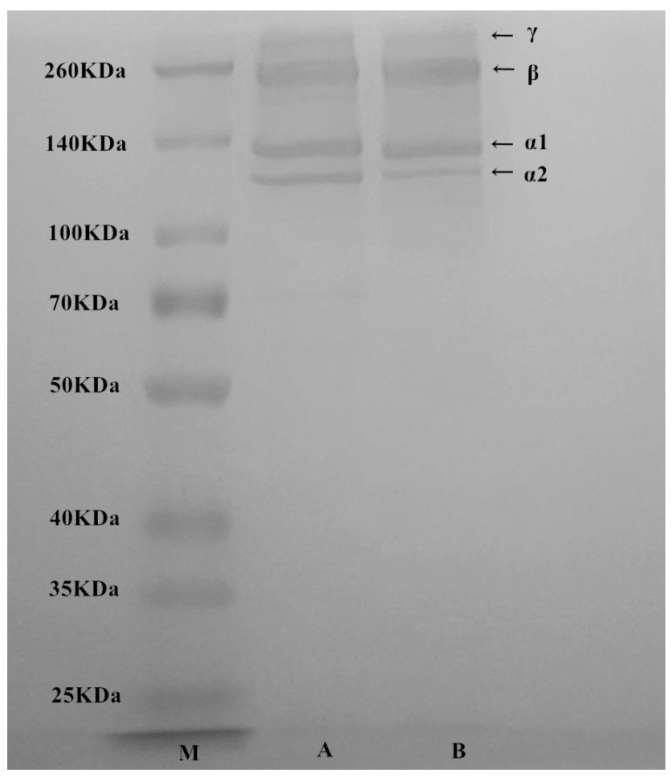
SDS-PAGE of molecular weight standard (lane M), authentic type І collagen standard from rat tail (lane A), and *T. flavidus* collagen extract (lane B).

**Figure 2 marinedrugs-17-00025-f002:**
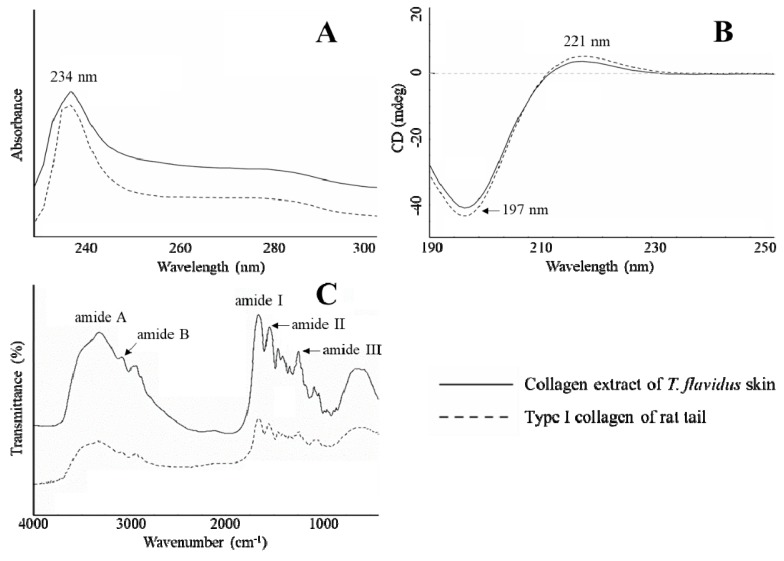
Spectrophotometric characterization of *T. flavidus* collagen extract and type І collagen of rat tail. (**A**) UV spectra; (**B**) circular dichroism (CD) spectra, and (**C**), FTIR spectra.

**Figure 3 marinedrugs-17-00025-f003:**
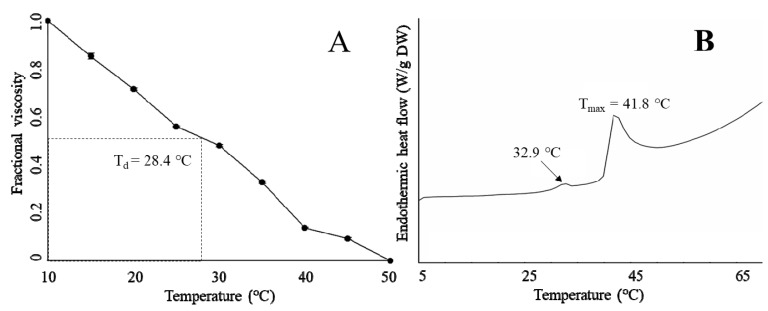
Fractional viscosity (**A**) and Differential scanning calorimetry (DSC) (**B**) of *T. flavidus* collagen extract.

**Figure 4 marinedrugs-17-00025-f004:**
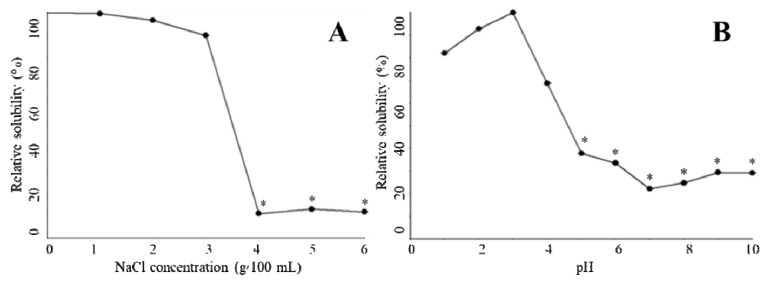
Solubility of *T. flavidus* collagen extract in the presence of varied NaCl concentration (**A**) or pH condition (**B**).

**Figure 5 marinedrugs-17-00025-f005:**
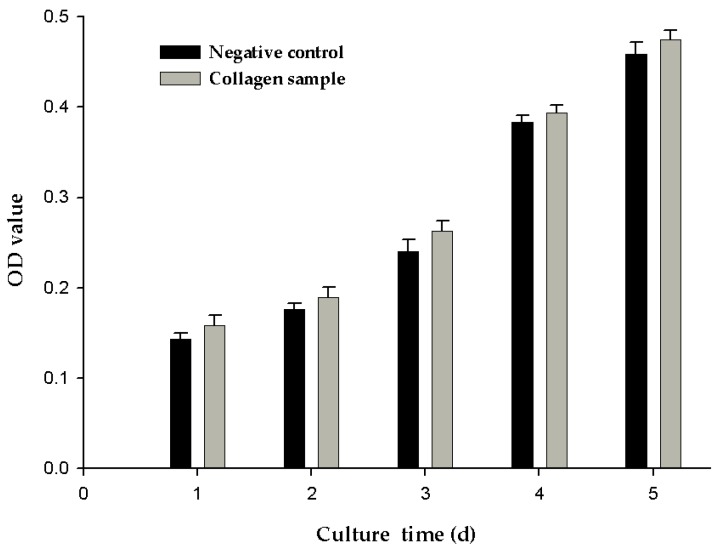
MTT assay detecting the effects of collagen sample on cell proliferation of NIH3T3 cell.

**Figure 6 marinedrugs-17-00025-f006:**
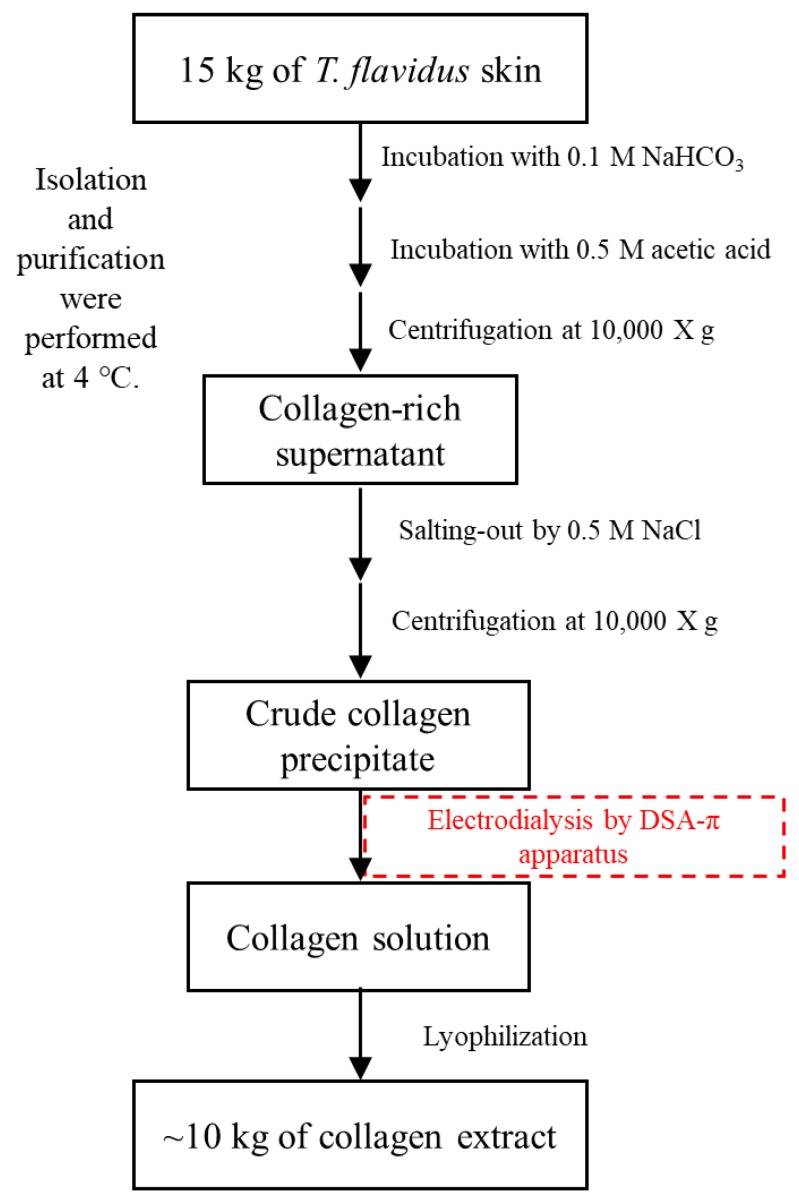
Isolation and purification of collagen from *T. flavidus* skin by electrodialysis.

**Table 1 marinedrugs-17-00025-t001:** Heavy metal concentrations (mg kg^−1^ dry weight).

Heavy Metal	Pufferfish Skin	Collagen Sample	Chinese National Standards ^a^	
			Food Additive Gelatin (GB 6783)	Maximum Concentrations of Contaminants in Foods (GB 2762)
Pb	0.562 ± 0.052	0.421 ± 0.015	≤1.5	≤0.5
Cd	0.042 ± 0.004	0.023 ± 0.002	-	≤0.1
As	0.075 ± 0.002	0.052 ± 0.001	≤1.0	≤0.1
Cr	1.152 ± 0.067	0.863 ± 0.072	≤2.0	≤2.0

^a^ Chinese national standards were obtained from the literature described by Chen et al. (2016).

**Table 2 marinedrugs-17-00025-t002:** Amino acid compositions of *T. flavidus* collagen (residues/1000 amino acid residues).

Amino acids	*T. flavidus*
Glycine	268 ± 0.08
Proline	164 ± 0.05
Hydroxyproline	82.1 ± 0.04
Arginine	32.7 ± 0.12
Hydroxylysine	9.2 ± 0.09
Lysine	37.7 ± 0.08
Alanine	119 ± 0.11
Threonine	39.8 ± 0.03
Valine	31.0 ± 0.09
Serine	14.7 ± 0.13
Isoleucine	13.9 ± 0.07
Leucine	33.8 ± 0.05
Methionine	18.5 ± 0.07
Histidine	13.7 ± 0.06
Phenylalanine	18.1 ± 0.03
Glutamine acid	37.1 ± 0.06
Aspartic acid	60.4 ± 0.05
Cysteine	0.57 ± 0.09
Tyrosine	5.43 ± 0.05

**Table 3 marinedrugs-17-00025-t003:** Expenses and instrumental requirements for producing 10 kg of *T. flavidus* collagen by electrodialysis compared to traditional dialysis.

	Traditional Dialysis	Electrodialysis
Essential Instruments	Dialysis bag, beakers and stirring hot plates	DSA-П electrodialysis apparatus
Instrument time (h/dialysis)	>96	2.0
Dialysis bag (m)	600	none
Acetic acid (mole)	18,000	45.0
Waste water (L)	360,000	600

Expenses and essential instruments for traditional dialysis was estimated based on the studies of Liu et al. (2012), Matmaroh et al. (2011), Nagai and Suzuki (2000), and Ogawa et al. (2004).

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
