# Peer review of "Electrodialysis Extraction of Pufferfish Skin (Takifugu flavidus): A Promising Source of Collagen"

_marinedrugs, 2019, doi:10.3390/md17010025_

Reviewer 1 Report

This is very interesting and well-written piece of applied science. Nevertheless, there are still some drawbacks that should be addressed.

1)    The choice of fish is unclear. Its geographical location is limited to Far East so it limits impact on the world industry, so more widely present fish would be a better option.

2)    The expense of energy and its cost has not been provided

3)    Potential contamination with toxic metals (as any marine product) and tetradotoxin (specific for fugu) has been checked.

4)    The word “Muslin” should be “Muslim”

5)    The word “desparately” should be removed from the introduction.

Author Response

Dear Reviewer,

I sincerely appreciate your comments of our work and they are very helpful for us. The manuscript has been revised according to your comments. All of the revisions made to our manuscript has been marked with red marker. Our responses for the comments were shown in attachment.

Thank you and best regards.

Yours sincerely,

Dr. Jun-De Chen,

Marine Biological Resource Comprehensive Utilization Engineering Research Center of the State Oceanic Administration, The Third Institute of Oceanography of the State Oceanography of the State Oceanic Administration, Xiamen 361005, China

Tel: +86-592-2195527;

Fax: +86-592-2195527

Reviewer 2 Report

The article is very interesting and fits in with the latest research trends. Research methodology and its implementation is very good, however, no information in Materials and Methods section  about collagen of rat tail, while the results we can see in Figs. 1 and 2.

Author Response

Dear reviewer,

We thank the reviewer for pointing out this important information and they are very helpful for us. We have added the origin of rat collagen in the method section: “Type I collagen (C3867, rat) was purchased from Sigma-Aldrich Inc. (MO, USA)”.

Thank you and best regards.

Yours sincerely,

Dr. Jun-De Chen,

Marine Biological Resource Comprehensive Utilization Engineering Research Center of the State Oceanic Administration, The Third Institute of Oceanography of the State Oceanography of the State Oceanic Administration, Xiamen 361005, China.

Tel: +86-592-2195527;

Fax: +86-592-2195527

Email: jdchen@tio.org.cn

Reviewer 3 Report

Line 36 – ‘cell grow’ needs to be corrected to ‘cell growth’

Line 47 – ‘Apart from the widely acceptance by different socio-cultural believes overall the world’ – this sentence doesn’t read well

Results and Discussion:

2.1 - Why was rat type 1 collagen chosen as a comparison if porcine and bovine are the most commonly used particularly in biomedical research? Would it not be more beneficial to compare against bovine or procine collagen?

I feel that this project would benefit from a cell study to determine the biocompatibility of the collagen and  if the ligands presented by type 1 collagen for cell attachment and growth are available on fish collagen extracted via electrodialysis. This is an important consideration, particularly as what makes collagen so attractive as a biomaterial and protein in research is its biocompatibility. A point highlighted by the authors in the third sentence of their introduction.

Author Response

Dear Reviewer,

I sincerely appreciate your comments of our work and they are very helpful for us. The manuscript has been revised according to your comments. All of the revisions made to our manuscript has been marked with red marker. Our responses for the comments were shown in attachment.

Thank you and best regards.

Yours sincerely,

Dr. Jun-De Chen,

Marine Biological Resource Comprehensive Utilization Engineering Research Center of the State Oceanic Administration, The Third Institute of Oceanography of the State Oceanography of the State Oceanic Administration, Xiamen 361005, China

Tel: +86-592-2195527;

Fax: +86-592-2195527

Round  2

Reviewer 1 Report

The authors added analysis of tetradotoxin and heavy metals content to the manuscript. However, since this fish is local and the authors referenced Chinese standards solely, I think the manuscript would be better suitable for local newspaper than the journal with worldwide reach.

There are also grammar errors in the newly added content such the one below:

“Heavy metals and tetrodotoxin analysis indicating that the security of T. flavidus collagen”

Reviewer 3 Report

I am happy with the changes made